# Prediction of cardiovascular disease risk among people with severe mental illness: A cohort study

Ruth Cunningham[1]*, Katrina Poppe[2], Debbie Peterson[1], Susanna Every-Palmer[3], Ian Soosay[4], Rod Jackson[2]

1 Department of Public Health, University of Otago Wellington, Wellington, New Zealand, 2 Section of Epidemiology and Biostatistics, School of Population Health, University of Auckland, Auckland, New Zealand, 3 Department of Psychological Medicine, University of Otago Wellington, Wellington, New Zealand, 4 School of Medical and Health Sciences, University of Auckland, Auckland, New Zealand

* Ruth.Cunningham@otago.ac.nz

## Abstract

### Objective

To determine whether contemporary sex-specific cardiovascular disease (CVD) risk prediction equations underestimate CVD risk in people with severe mental illness from the cohort in which the equations were derived.

### Methods

We identified people with severe mental illness using information on prior specialist mental health treatment. This group were identified from the PREDICT study, a prospective cohort study of 495,388 primary care patients aged 30 to 74 years without prior CVD that was recently used to derive new CVD risk prediction equations. CVD risk was calculated in participants with and without severe mental illness using the new equations and the predicted CVD risk was compared with observed risk in the two participant groups using survival methods.

### Results

28,734 people with a history of recent contact with specialist mental health services, including those without a diagnosis of a psychotic disorder, were identified in the PREDICT cohort. They had a higher observed rate of CVD events compared to those without such a history. The PREDICT equations underestimated the risk for this group, with a mean observed:predicted risk ratio of 1.29 in men and 1.64 in women. In contrast the PREDICT algorithm performed well for those without mental illness.

### Conclusions

Clinicians using CVD risk assessment tools that do not include severe mental illness as a predictor could by underestimating CVD risk by about one-third in men and two-thirds in

**Data Availability Statement:** Data used in this study are not freely available because of restrictions imposed by data providers and the ethical approval and research goals governing the study. Requests for data access would be subject

to scrutiny by researchers from the University of Auckland PREDICT research steering group and by Maori, Pacific and South Asian governance groups to ensure congruence with equity research goals. Applications will only be granted and data provided after agreement from our contributing providers and the Ministry of Health and after ethical approval by the New Zealand Mult-region Ethics Committee. For further enquiries, please contact Professor Rod Jackson (rt.jackson@auckland.ac.nz) or Dr Katrina Poppe (k.poppe@auckland.ac.nz), or the VIEW Governance Group Attn: Sally Gallaugher, School of Population Health, University of Auckland, Private Bag 92019, Auckland 1142, NZ, phone: +64 9 923 4888.

**Funding:** This study was supported by a grant from the National Heart Foundation of New Zealand, Project Grant reference 1685. All authors were named investigators on this grant application, and RC, DP and KP received funding from this grant. https://www.heartfoundation.org.nz/. The funders had no role in study design, data collection and analysis, decision to publish, or preparation of the manuscript.

**Competing interests:** The authors have declared that no competing interests exist.

**Abbreviations:** SMI, severe mental illness; CVD, cardiovascular diseases.

women in this patient group. All CVD risk prediction equations should be updated to include mental illness indicators.

## Introduction

Experience of severe mental illness (SMI) is associated with higher prevalence, incidence and mortality from a range of cardiovascular diseases (CVD) including coronary heart disease, congestive heart failure and cerebrovascular disease.[1] This increased risk of CVD is an important factor in the high rates of premature mortality among people with SMI.[1,2] SMI can be defined narrowly to include people with diagnoses of functional psychosis including schizophrenia and bipolar disorder, or more broadly to also include major depression and/or anxiety, or using a definition which relates to the level of need for services or functional disturbance, with evidence of increased CVD risk in all groups.[1–4]

Established risk factors such as smoking, diabetes, and obesity partly explain the increased CVD risk.[5,6] However, there is evidence that the increased risk exceeds that due to established risk factors.[7,8] Possible explanations include biological factors related to mental illness, under-recognition of CVD leading to delayed diagnosis, and lack of appropriate primary and secondary preventative interventions.[9]

Risk prediction algorithms such as the Framingham Risk Score [10] are important for informing appropriate management of primary CVD risk. If mental illness is an independent risk factor for CVD then risk prediction algorithms based on established risk factors will underestimate risk. Current UK NICE guidance on cardiovascular risk assessment recognises but does not quantify this likely underestimation in people with additional risk due to antipsychotic medication or SMI.[11] The new QRISK3 score used in the UK includes SMI and atypical antipsychotic prescription as predictors to rectify underestimation of risk.[12] However, other CVD risk assessment algorithms currently being used (for example PREDICT in New Zealand[13]) do not include SMI, and so empirical investigation is needed to understand the magnitude of underestimation.[14]

The main aim of our study was to compare the observed risk of a first CVD event among people with SMI with the risk predicted by recently developed algorithms for the New Zealand general population,[13] in order to identify and quantify any underestimation. The hypothesis was that CVD risk prediction algorithms will underestimate the CVD risk for this group. SMI was defined as treatment by specialist mental health services in the five years prior to CVD risk assessment. A subgroup with diagnoses of schizophrenia or bipolar disorder were examined separately to investigate the utility of using wider vs narrower definitions of SMI.

## Methods

### Study population

This study uses the PREDICT cohort, a population-based anonymised cohort of people having their first cardiovascular risk assessment in primary care in New Zealand. The PREDICT cohort has been well described elsewhere.[13,15] Briefly, the cohort includes all people who have their CVD risk assessed in primary care using the PREDICT-CVD web-based tool, which is used by 35 to 40% of New Zealand general practices covering approximately 35% of the national resident population, mainly in the Northern part of New Zealand. The cohort is continually being updated. For the analyses presented here, CVD risk assessments between 20 October 2004 and 31 December 2016 were included, with data from 522,969 individuals.

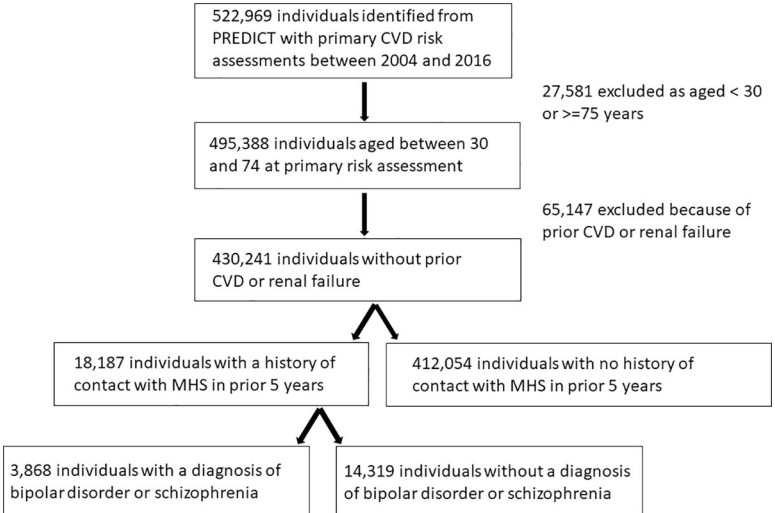

**Fig 1. Flow diagram for selection of cohort aged 30–74 without prior CVD from the PREDICT data set.**
Maximum follow up time was 12.2 years, and the mean follow up time was 4.5 years. Because comprehensive routine national data on deaths and hospitalisations was used for follow up there was no loss to follow up, except for people who left the country.

Routine CVD risk assessment was recommended from age 45 years (men) or 55 years (women) over the study period, and ten years earlier for people from Māori, Pacific and Indian subcontinent populations.[16] This cohort was linked to national pharmaceutical dispensing, hospitalisations, public mental health service records and mortality records using a unique national health identifier (the encrypted NHI). This study was limited to participants aged 30 to 74 years, which is the age group that the PREDICT CVD risk prediction algorithms were derived from. People with a history of prior CVD (including heart failure) or renal failure were excluded from the present study of primary risk prediction. Participants with missing data on predictor variables (2 with missing smoking status data values, 2659 with missing cholesterol data values) were excluded from the main analyses. Fig 1 details the cohort selection process.

## PREDICT cohort variables

**Exposure.** People with SMI were the primary exposure group of interest in the current study and this patient group has not been previously identified within the PREDICT Cohort. Data on treatment by specialist mental health services were used to identify the population group with the most functionally disruptive mental illness. Treatment by specialist mental health services in the five years prior to the index CVD risk assessment was identified from face-to-face treatment contacts with mental health services recorded in the PRIMHD (Programme for Integration of Mental Health Data) dataset, which covers all public secondary mental health care in New Zealand.[2] Inpatient and face-to-face community based mental health treatment contacts in the five years prior to the index assessment date were identified from information on the type and setting of service activities.

Mental health diagnoses were identified from the PRIMHD dataset. Missing diagnosis is a substantial problem with the PRIMHD dataset, with 36.2% of people identified in our study as having treatment contact with mental health services between 2009 and 2016 but having no psychiatric diagnosis information recorded in PRIMHD. All available psychiatric diagnoses, including secondary and provisional, were included to maximise available information. For the analyses presented here those with diagnoses of functional psychosis (ICD codes F20-F31)

were examined separately, as data on this diagnostic group are close to complete.[17] It was not possible to limit the overall definition of SMI to specific diagnoses because of missing diagnosis data, and so for this reason the whole group in contact with specialist services were defined as having SMI for the purposes of this study.

**Predictors used to calculate CVD risk.** Socioeconomic deprivation at the time of index assessment was measured using the New Zealand Deprivation Index (NZDep) 2013, which is an area-based measure of relative deprivation.[18]

We integrated multiple records of ethnicity from both the PREDICT dataset (recorded at primary care) and the National Health Index (recorded at secondary care) to ascertain a single prioritised ethnicity for each individual. The categories (in order of priority) are Māori (the indigenous population of New Zealand), Pacific, Indian, Chinese and other Asian, and residual group of other ethnicities (predominantly European).

The other predictors were standard CVD risk factors at baseline, which were drawn from data recorded by primary care clinicians at CVD risk assessment, augmented by prior hospitalisation, pharmaceutical dispensing, and lab test data. These risk factors were: age, gender, family history of premature cardiovascular disease, atrial fibrillation, diabetes, systolic blood pressure, TC:HDL ratio, and medications at index assessment (blood pressure lowering, lipid lowering, and antithrombotic medication). Definitions of these risk factor variables are available elsewhere.[16]

**Outcomes.** The primary outcome was total incident cardiovascular disease over the follow-up period, defined by ICD-10-AM codes as a death or hospitalisation from ischaemic heart disease, ischaemic or haemorrhagic cerebrovascular events, peripheral vascular disease, congestive heart failure or other ischaemic cardiovascular disease deaths.[13]

## Analysis

Descriptive analyses of demographics and risk factors were performed, stratified by sex and SMI. Numbers and proportions are reported to enable comparison between groups. All those with SMI were compared to those without SMI. The subgroup with a history of schizophrenia or bipolar disorder diagnosis are also described separately. Risk factors were described for those aged 30 to 74 years at index risk assessment, stratified by SMI and presented as numbers and proportions.

Time-to-event curves, adjusted for age were used to compare the risk of CVD outcomes (fatal or non-fatal) between those with SMI and those without. Those who died from a non-CVD cause were censored at date of death. Among the group with SMI, those with diagnoses of schizophrenia or bipolar disorder and those without were also examined separately.

Observed and predicted risk among people with SMI were compared using calibration plots. Calibration plots are also presented for those without SMI. The PREDICT algorithms, which have been developed and validated for the New Zealand population,[13] were used to calculate the predicted risk in deciles. Kaplan Meier estimates of observed risk were derived from CVD events in the five years following index CVD risk assessment. Men and women were examined separately. The ratio of predicted to observed risk was calculated for each decile of risk and the mean ratio reported to enable quantification of any underestimation.

We used SAS 9.4 and R software for analyses.

## Ethical approval

This study was approved by the New Zealand Northern A Health and Disability Ethics Committee, reference MEC/07/19/EXP/AM12. New Zealand ethics committees allow secondary re-use of health data without individual patient consent where data are not identifiable.

Information about the PREDICT study is available at all general practice locations, and patients may opt out of having their de-identified data being included in the cohort.

## Results

The PREDICT dataset was used to identify a cohort of 522,969 individuals who had a first CVD risk assessment between 2004 and 2016. This cohort was limited to 495,388 people aged 30 to 74 years at first risk assessment. Of these, 28,734 (5.8%) had also had face-to-face contact with specialist mental health services in the five years prior to their index CVD risk assessment, including 7669 (36.6%) women and 4,456 (15.5%) people with a recorded diagnosis of schizophrenia or bipolar disorder.

A total of 65,147 people were excluded from the analyses because of CVD or renal failure at the time of index risk assessment, leaving a final cohort of 430,241 individuals having a primary CVD risk assessment. Among women, 14.5% of those with SMI and 10.9% of those without SMI had prior CVD, while among men the proportions were 13.1% and 12.3% respectively.

Table 1 shows the demographic characteristics of the study participants aged 30 to 74 years without a prior history of CVD who had first CVD risk assessments over the study period. Those with SMI were younger at index assessment than those without SMI, and a higher proportion were Māori. SMI was associated with higher levels of deprivation, particularly among men and women with diagnoses of schizophrenia or bipolar disorder. In comparison to the total PRIMHD population, those who also appeared in the PREDICT dataset had a similar age and ethnicity distribution within the 45–74 year age group where routine screening was recommended (for example 49% aged 45–54 in PREDICT vs 53% in this age group in the total

**Table 1. Demographic factors at baseline CVD risk assessment among people 30–74 years with no prior CVD, by prior mental health (MH) status and gender.**

| | Women | | | | | | Men | | | | | |
| --- | --- | --- | --- | --- | --- | --- | --- | --- | --- | --- | --- | --- |
| | MH treatment past 5 years* | | Schizophrenia/ Bipolar Disorder | | No MH treatment past 5 years | | MH treatment past 5 years* | | Schizophrenia/ Bipolar Disorder | | No MH treatment past 5 years | |
| | n | % | n | % | n | % | n | % | n | % | n | % |
| *Total* | 6544 | | 1527 | | 181272 | | 11643 | | 2341 | | 230782 | |
| *Age (years)* | | | | | | | | | | | | |
| *30–44* | 903 | 13.8 | 257 | 16.8 | 12683 | 7.0 | 4227 | 36.3 | 962 | 41.1 | 49912 | 21.6 |
| *45–54* | 2603 | 39.8 | 620 | 40.6 | 56167 | 31.0 | 5052 | 43.4 | 915 | 39.1 | 93885 | 40.7 |
| *55–64* | 2310 | 35.3 | 495 | 32.4 | 77881 | 43.0 | 1776 | 15.3 | 346 | 14.8 | 57649 | 25.0 |
| *65–74* | 728 | 11.1 | 155 | 10.2 | 34541 | 19.1 | 588 | 5.1 | 118 | 5.0 | 29336 | 12.7 |
| *Ethnicity* | | | | | | | | | | | | |
| *Maori* | 1805 | 27.6 | 475 | 31.1 | 23857 | 13.2 | 3674 | 31.6 | 831 | 35.5 | 25911 | 11.2 |
| *Pacific* | 586 | 9.0 | 196 | 12.8 | 22886 | 12.6 | 1679 | 14.4 | 341 | 14.6 | 27841 | 12.1 |
| *Indian* | 281 | 4.3 | 67 | 4.4 | 14463 | 8.0 | 498 | 4.3 | 76 | 3.2 | 20855 | 9.0 |
| *Other Asian* | 374 | 5.7 | 80 | 5.2 | 20581 | 11.4 | 397 | 3.4 | 86 | 3.7 | 23435 | 10.2 |
| *European/other* | 3498 | 53.5 | 709 | 46.4 | 99485 | 54.9 | 5395 | 46.3 | 1007 | 43.0 | 132739 | 57.5 |
| *Deprivation Quintile* | | | | | | | | | | | | |
| *1 (least deprived)* | 1045 | 16.0 | 141 | 9.2 | 40416 | 22.3 | 1294 | 11.1 | 153 | 6.5 | 52754 | 22.9 |
| *2* | 1053 | 16.1 | 186 | 12.2 | 35966 | 19.8 | 1541 | 13.2 | 250 | 10.7 | 46675 | 20.2 |
| *3* | 1169 | 17.9 | 239 | 15.7 | 32972 | 18.2 | 1959 | 16.8 | 404 | 17.3 | 41960 | 18.2 |
| *4* | 1332 | 20.4 | 360 | 23.6 | 33598 | 18.5 | 2526 | 21.7 | 524 | 22.4 | 41915 | 18.2 |
| *5 (most deprived)* | 1945 | 29.7 | 601 | 39.4 | 38320 | 21.1 | 4323 | 37.1 | 1010 | 43.1 | 47478 | 20.6 |

* the numbers in this column include all those using mental health services including those with a schizophrenia or Bipolar disorder diagnosis

**Table 2. CVD risk factors at baseline CVD risk assessment and CVD events over follow up for people aged 30–74 years with no prior CVD, by prior mental health (MH) status and gender.**

| | Women | | | | | | Men | | | | | |
| --- | --- | --- | --- | --- | --- | --- | --- | --- | --- | --- | --- | --- |
| | MH treatment past 5 years* | | Schizophrenia/ Bipolar Disorder | | No MH treatment past 5 years | | MH treatment past 5 years* | | Schizophrenia/ Bipolar Disorder | | No MH treatment past 5 years | |
| | n | % | n | % | n | % | n | % | n | % | n | % |
| Total | 6544 | | 1527 | | 181272 | | 11643 | | 2341 | | 230782 | |
| Family history of CVD | 798 | 12.19 | 178 | 11.66 | 21433 | 11.82 | 1039 | 8.92 | 162 | 6.92 | 22682 | 9.83 |
| History of diabetes | 925 | 14.14 | 376 | 24.62 | 20952 | 11.56 | 978 | 8.40 | 401 | 17.13 | 21865 | 9.47 |
| History of atrial fibrillation | 52 | 0.79 | 10 | 0.65 | 1425 | 0.79 | 133 | 1.14 | 22 | 0.94 | 2777 | 1.20 |
| Lipid lowering medication | 920 | 14.06 | 300 | 19.65 | 28343 | 15.64 | 1295 | 11.12 | 423 | 18.07 | 34221 | 14.83 |
| BP lowering medication | 1471 | 22.48 | 364 | 23.84 | 47846 | 26.39 | 1626 | 13.97 | 365 | 15.59 | 44376 | 19.23 |
| Antithrombotic medication | 502 | 7.67 | 134 | 8.78 | 17031 | 9.40 | 675 | 5.80 | 162 | 6.92 | 20425 | 8.85 |
| Past smoker[a] | 1061 | 16.21 | 203 | 13.29 | 26780 | 14.77 | 1947 | 16.72 | 303 | 12.94 | 42742 | 18.52 |
| Current smoker | 1903 | 29.08 | 554 | 36.28 | 20974 | 11.57 | 4837 | 41.54 | 1145 | 48.91 | 34877 | 15.11 |
| Mean BMI[b] (kg/m$^2$) | 29.46 | (SD 7.44) | 31.67 | (SD 7.86) | 29.16 | (SD 7.18) | 29.51 | (SD 6.29) | 30.95 | (SD 7.17) | 29 | (SD 5.64) |
| BMI <25 | 1613 | 24.65 | 256 | 16.76 | 45785 | 25.26 | 2135 | 18.34 | 362 | 15.46 | 42185 | 18.28 |
| BMI 25–29 | 1547 | 23.64 | 342 | 22.40 | 44508 | 24.55 | 3417 | 29.35 | 623 | 26.61 | 78232 | 33.90 |
| BMI 30–34 | 1078 | 16.47 | 301 | 19.71 | 27651 | 15.25 | 2345 | 20.14 | 527 | 22.51 | 42332 | 18.34 |
| BMI 35–39 | 599 | 9.15 | 199 | 13.03 | 14922 | 8.23 | 900 | 7.73 | 236 | 10.08 | 15438 | 6.69 |
| BMI 40+ | 476 | 7.27 | 191 | 12.51 | 11815 | 6.52 | 571 | 4.90 | 204 | 8.71 | 8116 | 3.52 |
| Missing | 1231 | 18.81 | 238 | 15.59 | 36591 | 20.19 | 2275 | 19.54 | 389 | 16.62 | 44479 | 19.27 |
| Mean TC:HDL[c] | 3.86 | (SD 1.23) | 4.14 | (SD 1.37) | 3.71 | (SD 1.08) | 4.45 | (SD 1.42) | 4.79 | (SD 1.56) | 4.39 | (SD 1.24) |
| TC:HDL >4 | 2389 | 36.51 | 689 | 45.12 | 58941 | 32.52 | 6670 | 57.29 | 1524 | 65.10 | 131330 | 56.91 |
| Mean SBP, mmHg | 125.65 | (SD 17.55) | 123.92 | (SD 16.79) | 128.62 | (SD 17.59) | 127.1 | (SD 16.27) | 124.36 | (SD 15.81) | 128.82 | (SD 16.11) |
| Mean DBP, mmHg | 78.06 | (SD 10.76) | 77.7 | (SD 10.46) | 78.4 | (SD 10.15) | 80.09 | (SD 10.89) | 78.97 | (SD 10.42) | 80.07 | (SD 10.23) |
| Elevated BP[d] | 3528 | 53.91 | 756 | 49.51 | 109880 | 60.62 | 6739 | 57.88 | 1211 | 51.73 | 143798 | 62.31 |
| CVD events over follow up | 241 | 3.68 | 68 | 4.45 | 5880 | 3.24 | 448 | 3.85 | 88 | 3.76 | 9883 | 4.28 |

[a] Smoking status missing data values on 2 patients

[b] BMI missing data values for 84,576 patients (19.7%)

[c] TC:HDL ratio missing values for 3388 patients (0.8%)

[d] SBP>120 mmHg or DBP>90 mmHg

* the numbers in this column include all those using mental health services including those with a schizophrenia or Bipolar disorder diagnosis

PRIMHD population aged 45–74; 26% Māori in PREDICT vs 24% in PRIMHD). There was no missing data on demographic variables.

Table 2 shows the distribution of cardiovascular risk factors and cardiovascular outcomes by SMI history for men and women. With the exception of smoking, there were no marked differences in risk factor distribution among those with SMI compared to those without. However, people who had a recorded diagnosis of schizophrenia or bipolar disorder had higher rates of diabetes, obesity and hypercholesterolaemia than those without a history of mental illness, and the differences were more marked among women.

Fig 2 shows age-adjusted estimates of the risk of a CVD event over the first 8 years of follow up time stratified by history of mental illness. Among both men and women, a history of SMI is associated with an increased risk of a cardiovascular event. This was the case for people with diagnoses of schizophrenia or bipolar disorder and also for others using specialist mental health services, with overlapping confidence intervals for the two subgroups (see Fig 3).

Figs 4 and 5 compare observed 5-year risk (x axis) with deciles of predicted 5-year risk (y axis) of CVD events in the total study population aged 30 to 74 years. Estimates sitting on the

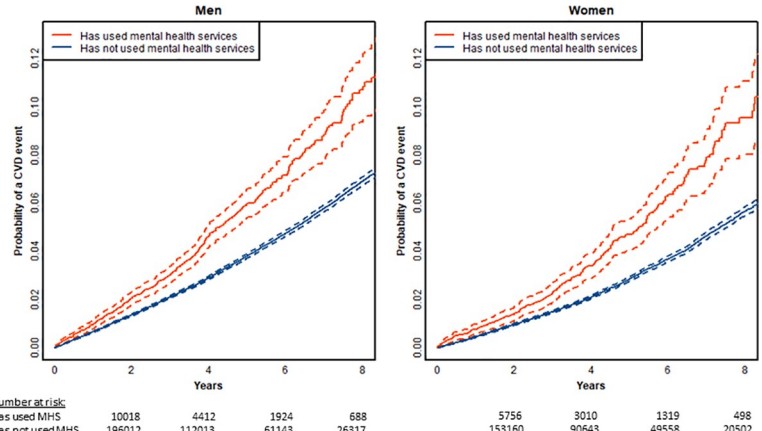

**Fig 2. Age adjusted time to event plots of risk of cardiovascular event by prior mental illness (SMI), limited to people aged 30–74 years at index assessment with no prior CVD, dashed lines indicate 95% confidence limits (n = 430241, events = 17197).**

diagonal line indicate no difference between observed and predicted risk per decile of the cohort (i.e. accurate risk prediction by the algorithm). Fig 4 compares people with SMI (left) and those without a history of mental health service use (right). For people with SMI, observed risk is higher than predicted risk across deciles of risk, indicating that the risk prediction algorithm is underpredicting the risk of CVD events. For example, in the highest decile of risk, the predicted risk of a CVD event was approximately 11% over 5 years, while the observed risk was approximately 14%. The same pattern was found in both men and women, although more pronounced among women (see Fig 5). The mean ratio of observed to predicted risk is 1.64 for women and 1.29 for men, and for men and women combined is 1.37. For those without a history of mental health service use the observed and predicted risks are approximately equal.

## Discussion

We found that among people aged 30 to 74 years without a history of CVD, who had a cardiovascular risk assessment in primary care, those with a history of SMI tended to be younger,

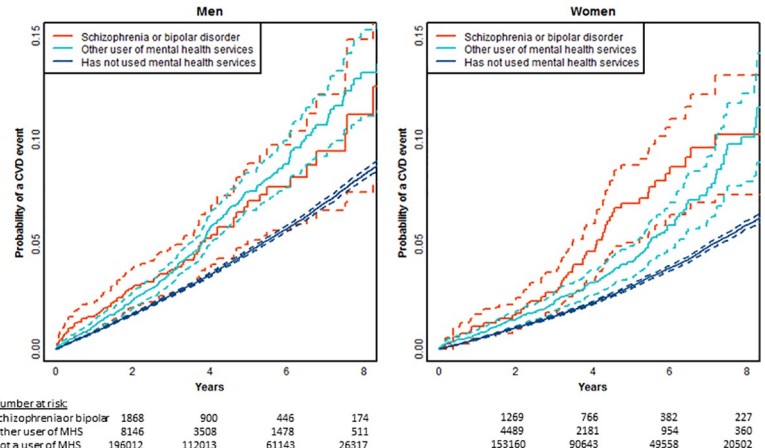

**Fig 3. Age adjusted time to event plots of risk of cardiovascular event across three categories of prior mental illness (SMI), limited to people aged 30–74 years at index assessment with no prior CVD, dashed lines indicate 95% confidence limits (n = 430241, events = 17197).**

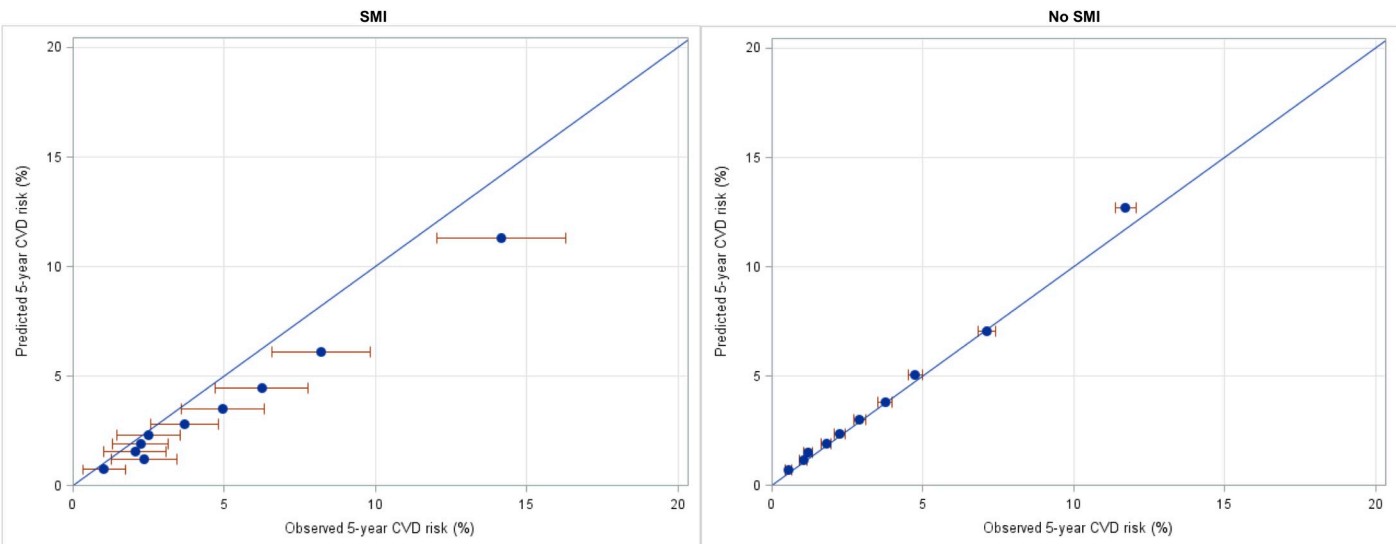

**Fig 4.** Predicted vs observed CVD risk among people aged 30–74 years with no prior CVD, with SMI (left), and without SMI (right) Error bars indicate 95% confidence limits. Blue diagonal line indicates observed = predicted risk (n = 426911, events = 16289).

more likely to be Māori, and live in more deprived areas. They also had higher smoking rates, although other risk factors were similar, except among the subgroup with schizophrenia or bipolar disorder who had higher rates of metabolic disturbances. The age-adjusted risk of CVD events was elevated in those with SMI, among those both with and without diagnoses of schizophrenia or bipolar disorder. When CVD risk predicted by the contemporary PREDICT algorithm was compared to the observed risk over five years, the algorithm consistently underestimated observed risk among both men and women with SMI, particularly in the top five deciles of predicted risk.

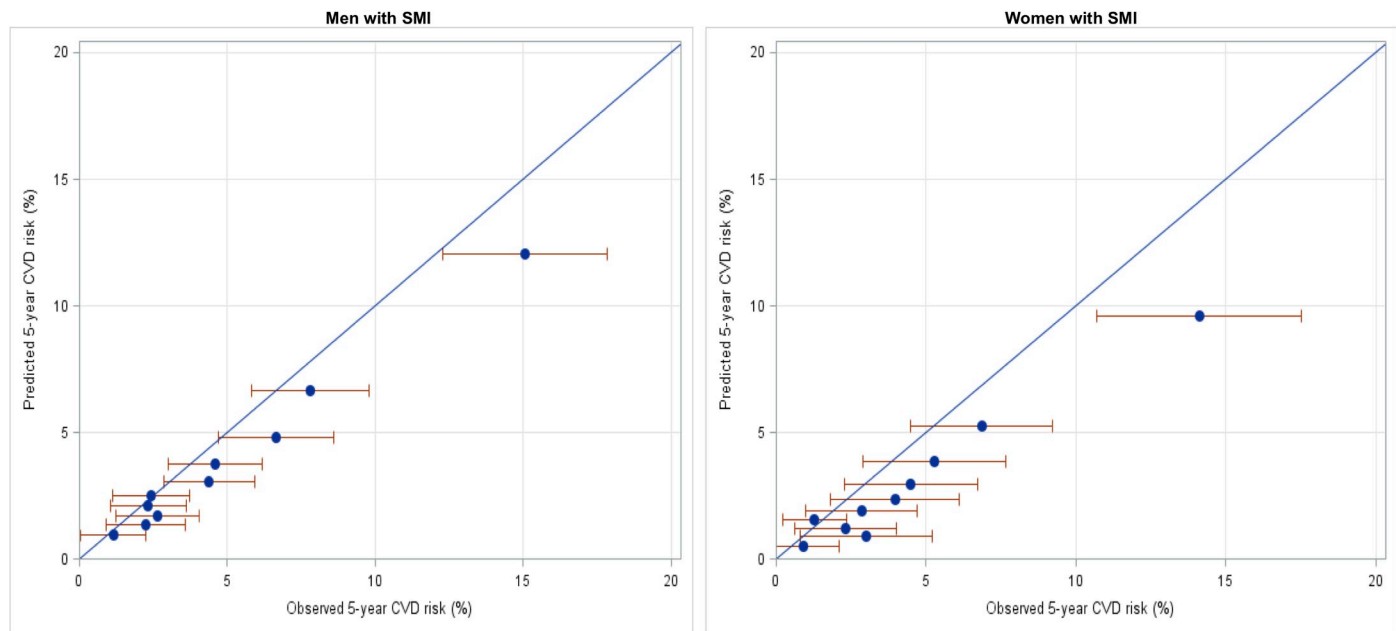

**Fig 5.** Predicted vs observed CVD risk among men (left) and women (right) with history of prior mental illness (SMI) in the five years prior to index assessment. Error bars indicate 95% confidence limits. Blue diagonal line indicates observed = predicted risk (n = 18055, events = 680).

These findings demonstrate that risk factor patterns including metabolic disturbances may only be part of the reason for the high rates of CVD among people with SMI and point to other factors such as overshadowing of physical health problems by mental health problems leading to delayed diagnosis and differences in the quality of preventive care received by those with mental illness compared to those without [4,9]. These mechanisms are likely to operate more strongly for more severe and stigmatised diagnoses and may partly explain the differences between those with and without functional psychosis.

## Strengths and limitations

We used a large primary care database with near complete coverage of the eligible population in the region in general practices using the PREDICT software, linked to data on specialist mental health service treatment to identify a history of SMI. Public specialist mental health services in New Zealand provide inpatient and community treatment for the approximately 3.5% of the population with the highest mental health need, making this an appropriate method for identifying a cohort with the most functionally disruptive mental illness.[19] People with diagnoses of functional psychosis or severe depression who have used specialist services more than five years previously or are being cared for by private services or primary care will not be identified as having SMI in this study. However, there is little private treatment for severe mental illness in New Zealand, and most people unwell with severe disorders use specialist care regularly, meaning that coverage of those with the most severe disorders will be reasonably complete. Those who are missed are likely to have less severe or disruptive conditions and so this method may overestimate the difference in cardiovascular risk between those with and without SMI if more severe illness is associated with a higher risk of CVD (as is suggested by the higher risk among those with psychosis diagnoses in this study). On the other hand, this method of identifying SMI will also include people with diagnoses not generally included in definitions of SMI, as well as people who have a relatively short duration of mental illness, and these groups may have a lower CVD risk than those with severe depression and psychosis, leading to a potential underestimate of the difference between those with and without SMI. Hence our method of ascertainment is imperfect but is likely to identify an unbiased population of those with SMI. Moreover, the separate analysis of those with psychosis provides a more precise and reproducible narrow definition of SMI. New Zealand has a similar patterns of both cardiovacular disease and mental illness to other high income countries and so these results are likely to be generalisable.

We only included people who had CVD risk assessed in primary care, and therefore run the risk of recruitment bias. Overall, 90% of eligible patients in the Auckland and Northland regions are included in PREDICT,[15] but it is not known whether a similar proportion of those with SMI are included. International data suggests lower rates of CVD risk assessment among those with SMI than the general population.[20] Moreover, a lack of clear demarcation of roles mean that some of those accessing secondary mental health services will be having their CVD risk assessed and managed by mental health services. Nevertheless, those who are not included in the study because they are missing out on CVD risk assessment or accessing CVD risk assessment elsewhere are likely to be at higher risk than those captured in routine primary care risk assessments, and so any bias would be in the direction of underestimating the difference in risk between those with and without mental illness.

One of the major strengths of the PREDICT cohort is the near complete availability of cardiovascular risk factor and demographic information for the cohort from primary care data, and of cardiovascular events through linkage to national hospitalisation and mortality data. Completeness of demographic and risk factor data was equal between those with and without

prior mental illness. It is possible that there is differential under-ascertainment of cardiovascular events among those with severe mental illness due to diagnostic overshadowing,[21] which would result in an underestimate of the risk of non-fatal cardiovascular events. Again, this bias would result in an underestimation of the difference between those with and without mental illness.

It is likely that given the extent of missing psychiatric diagnosis data, a small proportion of those without any recorded diagnosis did in fact have a diagnosis of functional psychosis. However, this was minimised by the inclusion of all provisional and primary psychosis diagnoses, including those recorded after the index assessment. Any misclassification would result in underestimating the difference between the two subgroups with SMI, but would not affect the main analyses which combined these groups.

## Strengths and weaknesses in relation to other studies

Other studies have compared predicted CVD risk between those with and without SMI, with many finding little difference in predicted risk between groups.[22–24] Although these studies point to underestimation, they were not able to confirm this as they did not include outcomes data. In contrast, our study included individual level outcome data so it was possible to investigate the performance of the algorithm to confirm and quantify underestimation. Two previous studies have also used both risk factor and outcome data for this population to understand increased risk and are discussed below.

Osborn and colleagues directly compared observed and predicted risk and found that the Framingham algorithm overestimated the risk of CVD events for people with SMI (as it does for the general population), and that Framingham recalibrated for the UK population under-predicted risk in women but not men with SMI.[25] This is consistent with our finding of a higher ratio of observed to predicted risk among women with SMI than men. However, this study did not include a comparison population without SMI, and did not use an up to date algorithm which would be recommend in practice.

In developing the UK based QRISK3 risk assessment algorithm, investigators found that both SMI (defined as schizophrenia or bipolar disorder) and atypical antipsychotic prescriptions independently predicted CVD outcomes, with an adjusted hazard ratio of 1.13 (men and women) for SMI and 1.29 (women) and 1.14 (men) for antipsychotics. Taken together, the magnitude of increased risk attributed to these two factors is comparable with the ratio of observed to predicted risk found in our study, despite the different definition of SMI used. Of particular note is the greater degree of increased risk among women with a history of mental illness compared to men with a similar history, which again was consistent with our findings.

A first cardiovascular risk assessment appeared to be done at a younger age among those with a history of SMI, although the pattern seen is also likely to relate to the young age distribution of those in contact with mental health services. Within the 45 to 74 years age range there was no evidence of earlier risk assessment among those with SMI, with the age distribution of people in PRIMHD and PREDICT mirroring the age distribution in PRIMHD. The recently updated New Zealand CVD consensus statement [26] recommends that people with SMI (defined as schizophrenia, bipolar affective disorder or major depression) have their CVD risk assessed regularly from the age of 25. This is an earlier age than those included in this study, and further work is needed to assess the impact and predictive power of risk assessment from this age.

We have confirmed that even the most up to date and well calibrated CVD risk algorithm for primary prevention substantially underestimates the risk for those with SMI. Importantly, this underestimation is not limited to those with a diagnosis of functional psychosis, who have

been the focus of previous studies.[25] It is therefore appropriate that those with major depression should be included in the definition of SMI for the purposes of identifying those at increased risk of CVD, as proposed in the NZ CVD consensus statement.[26]

## Conclusions

### Implications for practice

Demonstrating the magnitude of this underestimation of CVD risk is important for primary care practice, as mental illness is not specifically included most available risk prediction algorithms. The observed risk of an event in five years was 60% higher than estimated by the algorithm for women and 30% higher than estimated for men. Therefore, primary care providers using most common CVD risk prediction algorithms to inform management decisions will need to adjust the calculated risk upwards.

### Unanswered questions and future research

This study provides a clear rationale for the development of CVD risk prediction algorithms that include predictors for people with SMI. The updated PREDICT algorithm is in the process of being made available for practitioners in New Zealand and this presents an opportunity to include either a separate model for SMI or SMI as an additional predictor. Further investigation is needed to understand the reasons for the higher risk of CVD over and above that predicted by established risk factors. In particular, modifiable factors such as diagnostic overshadowing and differential receipt of treatment, need to be investigated to inform interventions to improve outcomes.

## Acknowledgments

The authors would like to thank Billy Wu for data management for the PREDICT study. The authors also thank the staff and patients in the primary health-care organisations using PREDICT software who contributed to the study, the Ministry of Health, Pharmac and Health Alliance for providing access to national and regional health databases, and Enigma Solutions Ltd for developing and implementing the PREDICT software, for preparing the data for analyses, and for providing the encrypted national health identifiers required for anonymised data linkage.

## Author Contributions

**Conceptualization:** Ruth Cunningham, Debbie Peterson.

**Data curation:** Katrina Poppe.

**Formal analysis:** Ruth Cunningham, Katrina Poppe.

**Funding acquisition:** Ruth Cunningham.

**Methodology:** Katrina Poppe.

**Project administration:** Ruth Cunningham.

**Supervision:** Rod Jackson.

**Visualization:** Katrina Poppe.

**Writing – original draft:** Ruth Cunningham.

**Writing – review & editing:** Debbie Peterson, Susanna Every-Palmer, Ian Soosay, Rod Jackson.

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
