## [Decision Letter · Decision Letter 0]

1 Jul 2019

PONE-D-19-15070

Prediction of cardiovascular disease risk among people with severe mental illness: a cohort study

PLOS ONE

Dear Dr. Cunningham,

Thank you for submitting your manuscript to PLOS ONE. After careful consideration, we feel that it has merit but does not fully meet PLOS ONE’s publication criteria as it currently stands. Therefore, we invite you to submit a revised version of the manuscript that addresses the points raised during the review process.

The two reviewers addressed a number of major and minor concerns about your manuscript. Please revise your manuscript carefully.

We would appreciate receiving your revised manuscript by Aug 15 2019 11:59PM. To enhance the reproducibility of your results, we recommend that if applicable you deposit your laboratory protocols in protocols.io, where a protocol can be assigned its own identifier (DOI) such that it can be cited independently in the future. For instructions see: http://journals.plos.org/plosone/s/submission-guidelines#loc-laboratory-protocols

We look forward to receiving your revised manuscript.

Kind regards,

Kenji Hashimoto, PhD

Academic Editor

PLOS ONE

**Journal Requirements:**

2. Thank you for stating that your study was approved by the New Zealand Northern A Health and Disability Ethics Committee, reference MEC/07/19/EXP/AM12.

Please place this information in your methods section.

3. Thank you for stating that the dataset was anonymised. Please provide this information in your ethics statement in the manuscript and in the online submission form. Please also clarify whether the ethics committee waived the requirement for informed consent. If patients provided informed written consent to have data from their medical records used in research, please include this information.

**Comments to the Author**

1. Is the manuscript technically sound, and do the data support the conclusions?

Reviewer #1: Yes

Reviewer #2: Yes

2. Has the statistical analysis been performed appropriately and rigorously? 

Reviewer #1: Yes

Reviewer #2: Yes

3. Have the authors made all data underlying the findings in their manuscript fully available?

Reviewer #1: Yes

Reviewer #2: Yes

4. Is the manuscript presented in an intelligible fashion and written in standard English?

Reviewer #1: Yes

Reviewer #2: Yes

5. Review Comments to the Author

Reviewer #1: This is a well-written paper and an important topic. The authors leveraged unique data sources to investigate CVD risk in those with severe mental illnesses (SMI). There are some concerns of the manuscript that limit its impact in its current form, as discussed in detail below. The three major concerns are the lack of clarity on the study population, the definition of SMI, and the lack of a proposed mechanism of the association between SMI and CVD events. The total number of participants in the PREDICT study was not provided nor was the way the investigators ended up with the analytic study population. It would help to show a diagram of the data sources and how the participants flowed into the final study population. The methods on page 4, line 84 state that data from 522,969 participants were included, but the tables show ~434,109. The other concern is with the ascertainment and definition of SMI. Could the authors comment on how accurate contact with specialist mental health services correlate to a SMI? It could be possible that people have depression or another SMI, but do not seek help from a specialist. Additionally, the definition of SMI is general and the diagnoses are missing. Both of these concerns could lead to misclassification of the exposure. If the authors conclude that SMI should be incorporated in CVD risk prediction tools, what variable should the tools use given the lack of SMI definitions in this analysis. Lastly, the paper would benefit from a proposed mechanism, which could vary depending on the specific SMI.

Minor areas of improvement for the authors to consider that could strengthen the paper are discussed below.

1. Do the authors compare the observed risk using the QRISK3 score that includes SMI?

2. Was there perfect linkage between the data sources?

3. The authors mention there was no loss to follow up, except for those who left the country. Could the authors provide the N?

4. How many events occurred and what are the sample sizes for the estimates in the figures?

5. Figure 1 shows the patterns of CVD events over 8 years. What is the rationale for showing 8 years?

6. Including those who had contact with a specialist mental health service, but did not have a diagnosis of a psychotic disorder is confusing. As mentioned, it is due to lack of data, but labeling those individuals as having a SMI is problematic.

Reviewer #2: This is a valuable article investigating the prediction of cardiovascular disease risk among people with severe mental illness. The paper reports that the PREDICT equations underestimated the risk for this group, with a mean observed: predicted risk ratio of 1.29 in men and 1.64 in women. This is a valuable paper that presents evidence of underestimating risk of CVD risk assessment tools in patients with severe mental illness.

I have the following concerns.

1. Discussion. P15, line 263. “hazard ratio of 1.[13]”

I think square brackets are unnecessary.

2. Discussion.

Whereas updated QRISK3 considers atypical antipsychotics and severe mental illness as risk factors, QRISK2 did not include them. I want to know if New Zealand people have plans to upgrade the PREDIC algorithm in the near future. If so, the findings of this study will be a great help.

In conclusion, I enjoyed reading this paper. This is a valuable paper that presents evidence of underestimating risk of CVD risk assessment tools in patients with severe mental illness.

6. PLOS authors have the option to publish the peer review history of their article (what does this mean?). If published, this will include your full peer review and any attached files.

Reviewer #1: No

Reviewer #2: No

---

## [Author Response · Author response to Decision Letter 0]

5 Aug 2019

Please note this response is also attached as a separate document

Response to reviewers:

Prediction of cardiovascular disease risk among people with severe mental illness: a cohort study

Thank you for the opportunity to submit a revised version of our manuscript and respond to the reviewers comments.

We have revised the manuscript and attach a tracked changes and clean copy of the revised manuscript.

We have included the requested extra information on the ethical approval process in the methods section of the paper and performed the requested formatting changes.

Below we have responded specifically to the comments from the reviewers (responses in red). Note: the line numbers refer to the tracked changes version of the revised paper.

Reviewer #1: This is a well-written paper and an important topic. The authors leveraged unique data sources to investigate CVD risk in those with severe mental illnesses (SMI). There are some concerns of the manuscript that limit its impact in its current form, as discussed in detail below. The three major concerns are the lack of clarity on the study population, the definition of SMI, and the lack of a proposed mechanism of the association between SMI and CVD events. 

The total number of participants in the PREDICT study was not provided nor was the way the investigators ended up with the analytic study population. It would help to show a diagram of the data sources and how the participants flowed into the final study population. The methods on page 4, line 84 state that data from 522,969 participants were included, but the tables show ~434,109.

Response:

We have amended the first paragraphs of the methods and results sections to make it clear that the initial sample (n=522,969) was restricted by age (n=495,388) and then further restricted by removing those with prior CVD to produce the final analytic sample (n=430,241). Note that the group with schizophrenia and bipolar disorder are a subset of the group who have had prior contact with mental health services, and so the total shown in table 1 and 2 is 430,241 not 434,109. We have added footnotes to the tables to clarify this. A flow diagram is also provided (new figure 1).

The other concern is with the ascertainment and definition of SMI. Could the authors comment on how accurate contact with specialist mental health services correlate to a SMI? It could be possible that people have depression or another SMI, but do not seek help from a specialist. Additionally, the definition of SMI is general and the diagnoses are missing. Both of these concerns could lead to misclassification of the exposure. If the authors conclude that SMI should be incorporated in CVD risk prediction tools, what variable should the tools use given the lack of SMI definitions in this analysis. 

Response:

Thank you for this comment. Our aim in this paper was to estimate the impact of mental illness on cardiovascular risk not only among those with a psychosis diagnosis but also among others with severe and persistent mental illness. This is important because of the evidence that non-psychotic disorders such as severe depression are also associated with an increased risk of CVD (for example Correll 2017). 

We have used contact with specialist mental health services as an indicator of SMI. The publically funded secondary mental health service in New Zealand aim to provide care for the 3% of the population with the highest mental health needs. The threshold for entry is mental illness that is too significant and too complex to be managed in primary care such that it requires specialist input. Therefore almost all of those accessing secondary mental health services in New Zealand will have serious mental illness.

It is worth noting that unlike most other Western countries, New Zealand does not have a well-developed private psychiatry workforce. For example, there is only one small private (not for profit) inpatient facility in the country, but even in this facility 2/3 of the beds are referred and funded through the secondary mental health services.

However, as the reviewer correctly comments, using contact with secondary mental health services as the exposure will not detect all people with SMI, as those with depression or another SMI may not seek help at all, or may be managed in primary care. It will also include people who would not be captured by a different definition of SMI which relied on diagnosis or duration of illness.

This is a limitation of our study. The study population represents a large cohort of people with SMI, but will not include everyone in the country with SMI. We have included further comment on this limitation in our methods (line 117–9) and the discussion (line 253–61), as well as comment on what this means for incorporating markers on severe mental illness in any risk prediction tool (line 327-9).

Lastly, the paper would benefit from a proposed mechanism, which could vary depending on the specific SMI.

Response:

As discussed in the second paragraph of the introduction to the paper, a number of possible explanations have been proposed for the increased risk of CVD among those with severe mental illness over and above that due to established risk factors including biological factors related to mental illness, under-recognition of CVD leading to delayed diagnosis, and lack of appropriate primary and secondary preventative interventions. We have included comment about potential mechanisms in the discussion (line 236). 

Minor areas of improvement for the authors to consider that could strengthen the paper are discussed below.

1. Do the authors compare the observed risk using the QRISK3 score that includes SMI?

Response:

We are unable to compare the observed risk with the QRISK3 score in this paper. The QRISK3 algorithm includes a number of factors which we do have recorded for our cohort. These include history of migraines, systemic lupus erythematosus, and erectile dysfunction. 

2. Was there perfect linkage between the data sources?

Response:

The data are linked using the unique identifier (NHI) which is present on all health service records, and which over 98% of New Zealanders have. It is increasingly rare for an individual to have more than one NHI however when duplicate records are identified (in the course of healthcare provision or during Ministry of Health audit), one of their NHI numbers is deemed the primary and others become linked to that as secondary NHI numbers. Linkage across data sources therefore recognises where individuals may have more than one NHI, meaning the risk of imperfect linkage is extremely minimal. 

3. The authors mention there was no loss to follow up, except for those who left the country. Could the authors provide the N?

Response:

Emigration status is not recorded in national health records so we are unable to estimate how many in the cohort may have left the country during the follow-up period of these data. We know that almost 34,000 New Zealand citizens left New Zealand in the year ending July 2018. If emigration is evenly distributed nationally it would be expected that 1/3rd of this number resided in the area covered by this study. We do not know how many of these were in our age range of 30-74 years, nor how many are likely to have severe mental illness. 

4. How many events occurred and what are the sample sizes for the estimates in the figures?

Response:

We have included the event numbers and total number of observations for each figure in the figure captions, and in the figures for the survival plots.

5. Figure 1 shows the patterns of CVD events over 8 years. What is the rationale for showing 8 years?

Response:

Eight years represented the 75th centile of the distribution of duration. The mean was 4.5 and maximum was 12.2 years, and so the inclusion of the final four years of data did not add extra information as it represented few people, and yet limiting to 5 years, for example, would misrepresent the usable information we have.

6. Including those who had contact with a specialist mental health service, but did not have a diagnosis of a psychotic disorder is confusing. As mentioned, it is due to lack of data, but labelling those individuals as having a SMI is problematic.

Response:

As noted above, the reason for including those without a diagnosis of a psychotic disorder is in order to investigate the relationship between mental illness and cardiovascular risk prediction accuracy for the broader group with mental illness at increased risk of CVD. We have clarified this and the limitations of our approach in the discussion of the paper.

Reviewer #2: This is a valuable article investigating the prediction of cardiovascular disease risk among people with severe mental illness. The paper reports that the PREDICT equations underestimated the risk for this group, with a mean observed: predicted risk ratio of 1.29 in men and 1.64 in women. This is a valuable paper that presents evidence of underestimating risk of CVD risk assessment tools in patients with severe mental illness.

I have the following concerns.

1. Discussion. P15, line 263. “hazard ratio of 1.[13]”

I think square brackets are unnecessary.

Response:

This was an error and we have made this change.

2. Discussion.

Whereas updated QRISK3 considers atypical antipsychotics and severe mental illness as risk factors, QRISK2 did not include them. I want to know if New Zealand people have plans to upgrade the PREDIC algorithm in the near future. If so, the findings of this study will be a great help.

Response:

The PREDICT algorithm used here is in the process of being made available for clinicians to use. There are plans to provide an adapted PREDICT algorithm which takes account of the increased risk among people with SMI. Our group are in discussions with the providers of the PREDICT software about how best to do this. We have included a sentence about this in the paper conclusions (line 340).

In conclusion, I enjoyed reading this paper. This is a valuable paper that presents evidence of underestimating risk of CVD risk assessment tools in patients with severe mental illness.

Thank you for the opportunity to submit a revised version of this paper for consideration for publication.

Yours sincerely,

Ruth Cunningham

---

## [Decision Letter · Decision Letter 1]

9 Aug 2019

Prediction of cardiovascular disease risk among people with severe mental illness: a cohort study

PONE-D-19-15070R1

Dear Dr. Cunningham,

We are pleased to inform you that your manuscript has been judged scientifically suitable for publication and will be formally accepted for publication once it complies with all outstanding technical requirements.

With kind regards,

Kenji Hashimoto, PhD

Section Editor

PLOS ONE

Additional Editor Comments (optional):

Reviewers' comments:

Reviewer's Responses to Questions

**Comments to the Author**

1. If the authors have adequately addressed your comments raised in a previous round of review and you feel that this manuscript is now acceptable for publication, you may indicate that here to bypass the “Comments to the Author” section, enter your conflict of interest statement in the “Confidential to Editor” section, and submit your "Accept" recommendation.

Reviewer #1: All comments have been addressed

Reviewer #2: All comments have been addressed

2. Is the manuscript technically sound, and do the data support the conclusions?

Reviewer #1: Yes

Reviewer #2: Yes

3. Has the statistical analysis been performed appropriately and rigorously? 

Reviewer #1: Yes

Reviewer #2: Yes

4. Have the authors made all data underlying the findings in their manuscript fully available?

Reviewer #1: Yes

Reviewer #2: Yes

5. Is the manuscript presented in an intelligible fashion and written in standard English?

Reviewer #1: Yes

Reviewer #2: Yes

6. Review Comments to the Author

Reviewer #1: The authors were responsive to the reviewer comments. Since there was excellent data linkage, one suggestion would be to elaborate on it more as a strength on page 16, line 269.

Reviewer #2: I believe the paper will be of interest to the readership of Plos One and would recommend it for acceptance.

7. PLOS authors have the option to publish the peer review history of their article (what does this mean?). If published, this will include your full peer review and any attached files.

Reviewer #1: No

Reviewer #2: No

---

## [Editor Report · Acceptance letter]

27 Aug 2019

PONE-D-19-15070R1 

Prediction of cardiovascular disease risk among people with severe mental illness: a cohort study 

Dear Dr. Cunningham:

I am pleased to inform you that your manuscript has been deemed suitable for publication in PLOS ONE. Congratulations! Your manuscript is now with our production department. 

With kind regards,

on behalf of

Prof. Kenji Hashimoto 

Section Editor

PLOS ONE